# Exploration of the Effect of Oxygen on Superconductivity in MgB₂ Bulk by Using Boron Powder with Different Particle and Purification

**Liangqun Yang [1], Hongli Suo [1,\*], Lin Ma [1], Min Liu [1], Wanli Zhao [1], Jianhua Liu [2], Lei Wang [2], Zili Zhang [2,\*] and Qiuliang Wang [2,3]**

[1] Beijing University of Technology, Beijing 100124, China; yangliagnqun@emails.bjut.edu.cn (L.Y.); malin@bjut.edu.cn (L.M.); lm@bjut.edu.cn (M.L.); zhaowanli@mails.bjut.edu.cn (W.Z.)

[2] Institute of Electrical Engineering, Chinese Academy of Sciences, Beijing 100190, China; liujianhua@mail.iee.ac.cn (J.L.); wanglei@mail.iee.ac.cn (L.W.); wongsm17@mail.iee.ac.cn (Q.W.)

[3] University of Chinese Academy of Sciences, Beijing 100049, China

\* Correspondence: honglisuo@bjut.edu.cn (H.S.); zhangzili@mail.iee.ac.cn (Z.Z.)

**Abstract:** In this study, boron powder with different particle sizes was purified by both chemical and heat treatment methods. The reduction in the particle size can improve the chemical purification with no effort on the heat treatment. The superconducting properties of the powder drastically changed even with only a partial elimination of oxygen. On the one hand, less oxygen content resulted in high $T_c$ and $J_c$ values under the low magnetic field, and most importantly, a significant improvement in the superconducting connectivity ($A_f$ value). On the other hand, the degradation of $J_c$ under a high field and a change in the pinning mechanism were also found, along with decreasing oxygen. This result indicated that oxygen, probably MgO, might act as the pinning center and as an obstacle for the supercurrent in MgB₂ at the same time. This work paves the way for obtaining pure oxygen-free MgB₂ and understanding the real effect of oxygen in MgB₂.

**Keywords:** boron powder; purification; particle size; MgB₂; superconducting connectivity

## 1. Introduction

MgB₂, with a transition temperature of 39 K, was discovered in 2001 and has attracted significant attention worldwide [1] owing to its weak-link free-grain boundaries, considerable coherence length, low material cost, and superconducting properties. However, the presence of the impurity, MgO, is inevitable [2–5] in all kinds of MgB₂ materials, including bulk, wires, tapes, thin films, and single crystals.

The effect of MgO on MgB₂, especially in terms of its superconductivity, remains unclear after nearly 20 years of research. Some groups insist that MgO is one of the main barriers of the supercurrent flow, which decreases its critical current density ($J_c$) [6–8]. Some groups believe that MgO is a kind of dopant that can increase the pinning force and $J_c$ at the high magnetic field. For example, Jiang et al. added nanosized MgO to MgB₂ and found $J_c$ to be enhanced [9].

In summary, there are no reports on obtaining pure MgB₂ without any oxygen impurity, and thus, the effect of MgO on MgB₂ remains unclear. There are two MgO sources, namely, MgO existing in Mg precursor powder and B₂O₃ existing in B precursor powder. These powders react with Mg during synthesis. Only a few groups have tried to eliminate the oxygen in Mg powder [10,11].

Most studies on purification have focused on B powder. The first study on low-purity B powder on MgB₂ was reported in 2006 [12,13]. Since then, few researchers have purified B powder in order to eliminate the oxygen [12,14,15]. In 2019, we reported a systematical investigation on the purification of B powder by both heat treatment and chemical methods [16]. At that time, we found the heat treatment method to be more

effective than the chemical method; however, both methods had limitations. The oxygen in the B powder could not be eliminated. Based on our analysis, as shown in Figure 10 from Reference [16], when a small B particle agglomerates to a larger one, the oxygen inside the large particles cannot be purified.

In this study, we determined the effect of particle size on the purification results obtained using the heat treatment method and chemical method to test if our analysis is correct. The effect of the initial oxygen purification on the superconductivity was also investigated, which can offer more insights into the real effect of oxygen on the superconducting properties of $MgB_2$.

## 2. Materials and Methods

The purification process is described in our previous report [16]. Briefly, the B powder was treated by the high temperature and chemical solution. In this paper, we only chose the 15 vol% HCl solution. To explore the effect of particle size, for the sample with grinding, the B powder was manually ground in an agate mortar for 0.5 h and then purified by the aforementioned methods. Only the best-purified B powder and Raw B powder were used to synthesize $MgB_2$ bulk using the same process as our previous report [16].

An oxygen and nitrogen analyzer (Mettler GPro 500, Udorf, Switzerland) was used to determine the oxygen content. The phase analysis of the samples was conducted using X-ray diffraction (XRD, Bruker D8, Karlsruhe, Germany) with Cu Kα. Lattice parameters, the grain size, and the phase proportion were calculated from the XRD results using the Rietveld method. The microstructures of the samples were observed by scanning electron microscopy (SEM, FEI Quanta FEG, Hillsboro, OR, USA) and transmission electron microscopy (TEM, Tecnai G2 20 S-TWIN FEI, Hillsboro, OR, USA). The zero-field-cooled curve was measured using a physical property measurement system (PPMS, Quantum Design, San Diego, CA, USA) under an applied magnetic field of 5 mT. The critical current densities, $Jc$ (in Am$-2$), of the samples were calculated by applying a standard Bean model expression for spherical grains to magnetic hysteresis loops using the following formula:

$$Jc = \frac{\Delta M}{b(1 - b/3a)}$$

where $\Delta M$ (in Am$-1$) is the vertical width of the magnetization loop, and $a \geq b >> c$ (in m) are the dimensions of the measured bulk. Resistance vs. temperature curves (R-T curves) under different applied fields of $MgB_2$ bulks were recorded. In the curve, 10% and 90% of transition points are defined as the relevant irreversible field ($H_{irr}$) and upper critical field ($H_{c2}$).

## 3. Results

Figure 1 shows the XRD results of samples prepared using B powders with different particle sizes. The $B_2O_3$ peak is significantly eliminated in all purified samples, as evidenced by the dashed line, which is consistent with the result reported by Jiang et al. [11]. Table 1 shows the oxygen contents and particle sizes of different B powders. After grinding, the average particle size was decreased. As shown in Figure 2, the large chunk of B powder was separated into a smaller one, which caused more surface of the B powder to be exposed. At the same time, the morphology of the different B powders was nearly the same. It allows us to investigate the effect of the particle size of B powder on the purification results solely and test our former speculations [10].

It is interesting to note that although both chemical and heat treatment methods could eliminate the oxygen content in the B powder, when the particle size was reduced, there were different results. In samples subjected to chemical purification, the oxygen content continuously decreased as the particle size was decreased. However, in samples subjected to heat treatment, the particle size did not affect the oxygen content. It implies that there is an intrinsic difference in the purification mechanism between the two methods. We will discuss this part in a later section.

The grinding+1050-purified B powder and unpurified sample were used in the synthesis to explore the effect of the purification on the superconducting properties of $MgB_2$. Figure 3a shows the XRD results of $MgB_2$ bulk containing B powders with different particle sizes. Both samples had $MgB_2$ as the main phase with a small amount of MgO. The area of 61° to 65° is zoomed in to focus on the MgO (002) peak and $MgB_2$ (102) peak, as shown in Figure 3b. As seen, the MgO peak intensity is higher than the $MgB_2$ peak intensity in the Raw B sample but nearly the same in the purified one. These results sufficiently prove the oxygen elimination effect of the purification process.

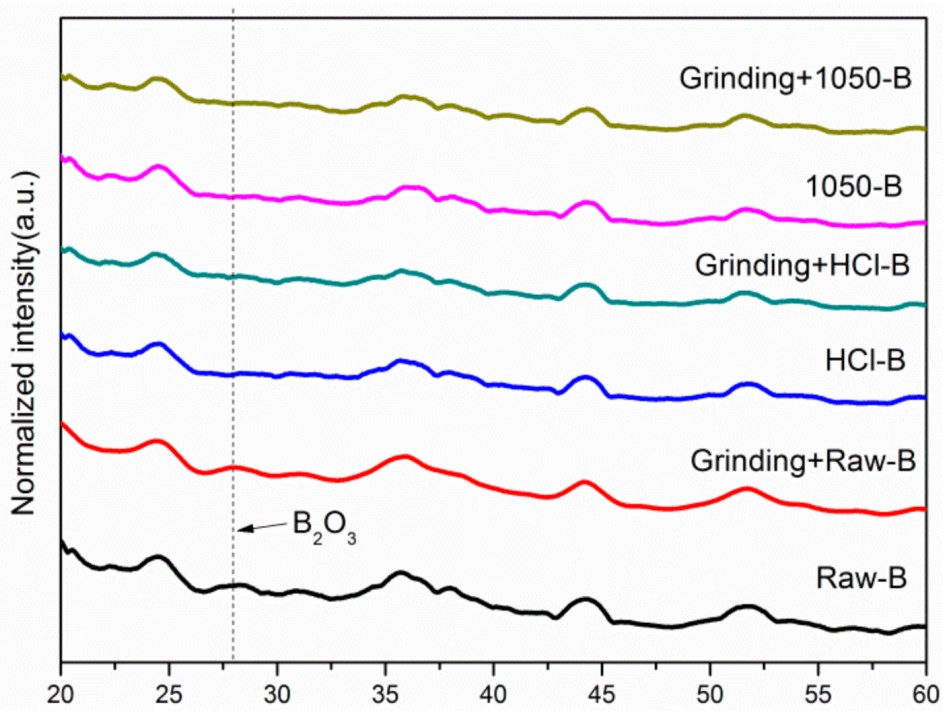

**Figure 1.** XRD results of B powder after different purifications.

**Table 1.** List of boron powders with different purification methods, relevant oxygen content, and particle size.

| Sample Name | Purification Method | Oxygen Content (ppm) | Particle Size | |
|---|---|---|---|---|
| | | | Average Size (μm) | Standard Deviation (%) |
| Raw B | Without any purification | 37,692 ± 112 | 0.5855 | 0.4217 |
| HCl B | Purified B powder by 15 vol% HCl solution | 22,408 ± 104 | 0.7064 | 0.6375 |
| Grinding+HCl B | Grinding powder and purified B powder by 15 vol% HCl solution | 16,022 ± 113 | 0.3407 | 0.5396 |
| 1050 B | Heat treatment at 1050 °C for 24 h | 9519 ± 120 | 0.7169 | 0.3479 |
| Grinding+1050 B | Grinding powder and heat treatment at 1050 °C for 24 h | 9532 ± 112 | 0.5636 | 0.4547 |

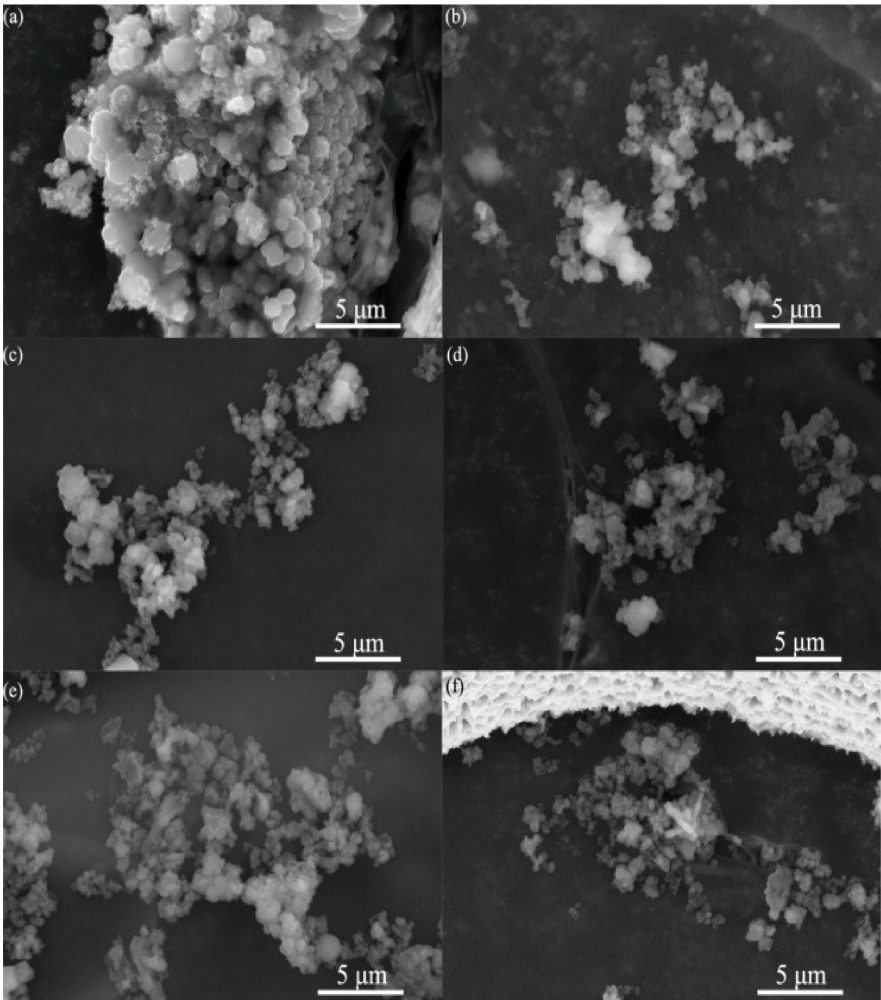

**Figure 2.** SEM images of the B powder purified with different methods. (**a**) Raw B powder; (**b**) B powder after grinding; (**c**) HCl-purified B powder; (**d**) Grinding+HCl-purified B powder; (**e**) heat treatment (1050 °C)-purified B powder; (**f**) Grinding+heat treatment (1050 °C)-purified B powder.

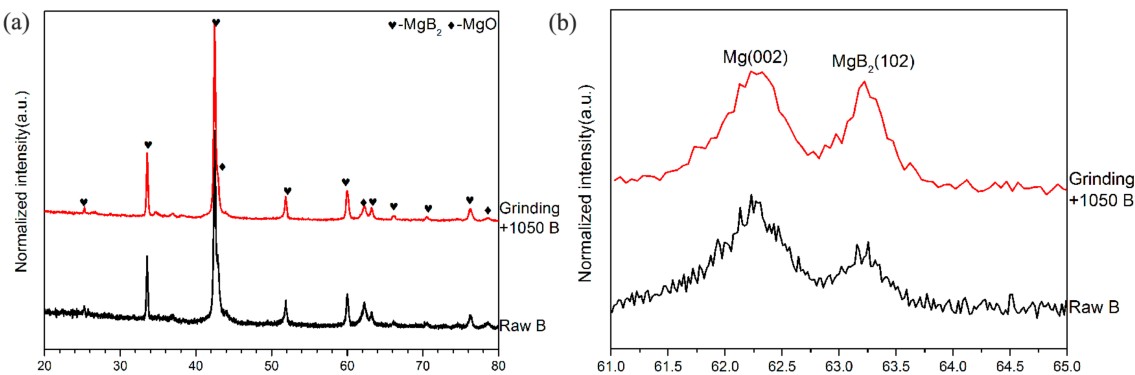

**Figure 3.** (**a**) XRD results of MgB$_2$ bulk prepared using B powder with different particle sizes; (**b**) The XRD data in the range between 61° and 65°.

Table 2 summarizes the calculated lattice parameters, the grain sizes, and the phase contents of MgB$_2$ bulk prepared using B powders with different particle sizes. The lattice parameters of both samples shown in Table 2 were nearly the same as their theoretical values ($a = 3.084$, $c = 3.522$), indicating that the purification of B powder did not affect the MgB$_2$ lattice structure. However, the grain size presented a noticeable difference. After

purification, the MgB$_2$ grains had evident growth. It is thought that the change in the grain size was mainly caused by the elimination of oxygen, which will be discussed later.

**Table 2.** Phase formation, lattice parameters, and grain sizes of the MgB$_2$ bulk prepared using B powder with different purification methods.

| Sample Name | Lattice Parameter (Å) | | Grain Size (nm) | MgB$_2$ (%) | MgO (%) |
|---|---|---|---|---|---|
| | a | c | | | |
| Raw B | 3.083 | 3.520 | 24.5 | 81.9 | 19.1 |
| Grinding+1050 B | 3.085 | 3.525 | 38.7 | 92.12 | 7.88 |

Figure 4 shows the microstructures of bulk MgB$_2$ prepared using different B powders. Under low magnification, both samples show identical porous structures without any visible difference. Under high magnification, in the unpurified sample, the boundary of each MgB$_2$ particle is clear. However, the MgB$_2$ particle in the purified sample shows better connectivity than that in the unpurified one. To obtain more details on the microstructure, TEM was used to observe individual particles. As seen in Figure 4e, there are many small dots with a size of tens of nanometers. In contrast, the MgB$_2$ particle in the purified sample is very clean. Thus, the nanosized dots in Figure 4e could be MgO.

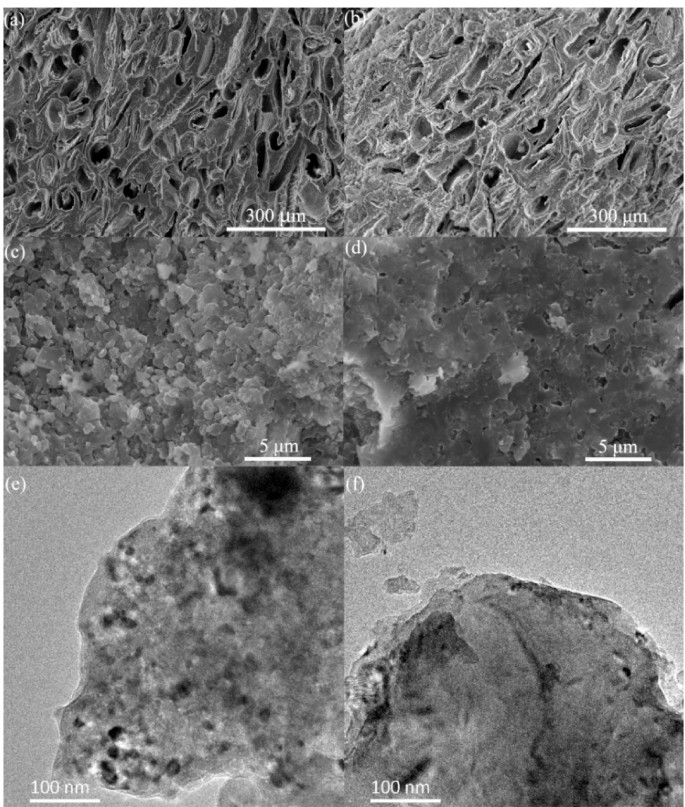

**Figure 4.** Microstructures of MgB$_2$ bulk prepared using B powders with different particle sizes (**a**) and (**c**): SEM images of Raw B; (**b**) and (**d**) SEM images of Grinding+1050 B; (**e**) TEM image of Raw B; (**f**) TEM image of Grinding+1050 B.

Figure 5 show the superconducting properties of the MgB$_2$ bulk prepared using different B powders with different purification. As shown in Figure 5a–c, the purified sample has a higher $J_c$ value under low field at all measured temperatures; however, it decreases faster than that corresponding to the unpurified one. Furthermore, such a cross point was found between the two $J_c$ curves around 3.5 T at 20 K, also proved by the pinning

force results, as shown in Figure 5d–f. Consequently, purification can only improve $J_c$ under low field but degrade the pinning of the $MgB_2$.

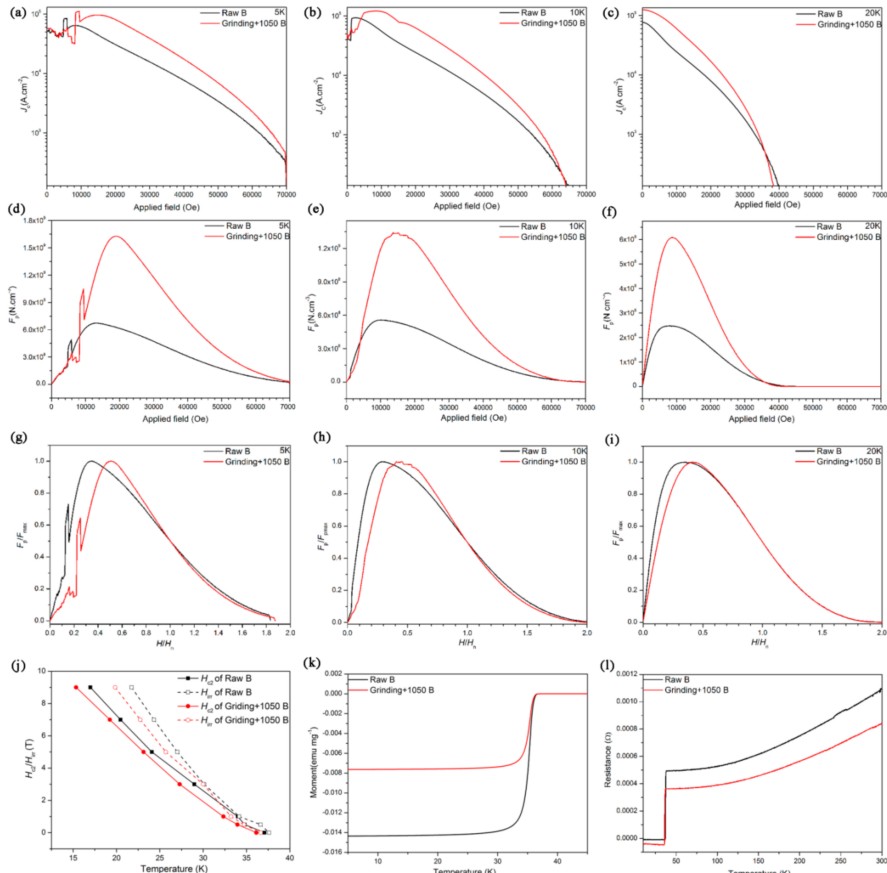

**Figure 5.** Superconducting properties of $MgB_2$ bulk with B powder of different sizes. (**a**) $J_c$ vs. applied field at 5 K; (**b**) flux pinning force ($F_p$) vs. applied field at 5 K; (**c**) $F_p/F_{p\,max}$ vs. $H/H_n$ curve at 5 K; (**d**) $J_c$ vs. applied field at 10 K; (**e**) flux pinning force ($F_p$) vs. applied field at 10 K; (**f**) $F_p/F_{p\,max}$ vs. $H/H_n$ curve at 10 K; (**g**) $J_c$ vs. applied field at 20 K; (**h**) flux pinning force ($F_p$) vs. applied field at 20 K; (**i**) $F_p/F_{p\,max}$ vs. $H/H_n$ at 20 K; (**j**) irreversible field ($H_{irr}$) and upper critical field ($H_{c2}$) vs. temperature curve; (**k**) magnetization vs. temperature curve; (**l**) resistance vs. temperature curve.

As there is an apparent difference in the pinning force, it is worth exploring the pinning mechanism further. The pinning force curves are fitted based on Ref. [17]. Moreover, to eliminate the effect of anisotropy in bulk $MgB_2$, $H_n$, where $F_p$ reaches half of $F_{p\,max}$, is used instead of $H_{irr}$ [18]. All the $F_p/F_{p\,max}$ curves are fitted using the simple scaling laws

$$F_p \propto h^m(1 - h^n),$$

where $h$ is the reduced field $h = H/H_n$ [18], as shown in Figure 5g–i. Furthermore, there is a noticeable difference in the pinning mechanism between the samples prepared using B powders with different particle sizes. The $m$ and $n$ values are 1/2 and 2 for the unpurified samples and 1 and 2 for the purified one. This fitting number combination corresponds to a standard surface pinning and normal point pinning. Subsequently, the pinning center becomes smaller as the B powder is purified.

Figure 5j shows the critical field of bulk $MgB_2$ prepared using B powders with different particle sizes. Although purification can make $MgB_2$ clearer, the critical field, especially $H_{c2}$, was degraded. Figure 5k shows the critical transition value. Both samples have a steep transition curve. Moreover, due to the better phase formation of the sample prepared using purified B powder, $T_c$ is slightly higher. One of the primary roles of MgO in $MgB_2$ is

the obstacle of supercurrent, so evaluating superconducting connectivity is very crucial. Rowell showed that $A_f$ value, where $A_f = \frac{\Delta \rho_{ideal}}{\Delta \rho} = \frac{\rho_{300K\ ideal} - \rho_{40K\ ideal}}{\rho_{300K} - \rho_{40K}}$, can quantitively evaluate the superconducting connectivity of bulk $MgB_2$ [19]. In this formula, $\rho_{300K}$ and $\rho_{40K}$ are the measured resistivities of bulk $MgB_2$, as shown in Figure 5l, and $\Delta \rho_{ideal}$ is chosen as 7.3 μΩ cm [20,21]. $A_f$ values of the sample with Raw B and purified B powder are 6.49% and 11.32%, respectively.

## 4. Discussion

Based on the results above, there is no doubt that the B powder's purification can affect the superconducting properties. However, there are still three questions that need to be discussed briefly.

*What is the difference between the chemical purification method and heat treatment purification method?*

Interestingly, reducing the particle size can further enhance the purification effect of the chemical method but does not affect the heat treatment method. It partially proved our speculation that there is some $B_2O_3$ inside the big chunk of B powder. This $B_2O_3$ barely touched the chemical solution. Subsequently, when the big chunk is broken, the $B_2O_3$ can be eliminated. However, during the heat treatment purification, the $B_2O_3$ also can melt and volatilize. Therefore, reducing the particle size did not cause any positive effect.

After multiple research studies on B powder's purification by using both chemical method and heat treatment method, the advantages and disadvantages of each method are summarized. The purification effect is not strong enough for the chemical method, especially for the micro size B power. Decreasing B powder's particle size can help eliminate the oxygen better, but it is still not good enough. According to the literature [22,23], the hydroxyl in the chemical solution could attach to the surface of the boron powder, which can cause further oxidization during the synthesis of $MgB_2$. In contrast, the purification effect of the heat treatment method is much better, and it does not have the hydroxyl problem. However, the high temperature is a highly sensitive environment for B powder. Even with a small amount of oxygen, it can cause severe oxidation of B powder. Thus, a special furnace or furnace tube is required. Furthermore, until now, we did not find any further agglomeration of B powder after a long heat treatment time. However, the B powder is microsize right now, which does not tend to agglomerate to each other. If the nanosize B powder is used, there is a possibility that the particle size will increase after heat treatment, which may affect the superconducting properties of $MgB_2$.

*Why could we not completely eliminate the oxygen in B powder?*

We now investigate why we could not wholly purify all the oxygen in B powder. There are two possible reasons: where is the oxygen, and what is the oxygen?

As shown in the previous research [16], it is thought that the oxygen was embedded inside the B powder, which could not be eliminated. In this study, we tried to grind the B powder to decrease the particle size, which can expose more oxygen inside the powder. It is found that such a process only worked on the chemical solution method. It proved our previous speculation. Since the particle size is only partially decreased, which is far away from B powder's grain size, there is a possibility that the residual oxygen still existed inside the powder.

Since all our purification processes are focused towards eliminating the $B_2O_3$, if the oxygen content is not $B_2O_3$, our process will never work. As such, the second reason might be relevant to how the commercial powder was a Self-propagating high-temperature synthesis [24]. The gas route gives very pure crystalline boron ($\geq$99%) in standard conditions, whereas the magnesiothermic reaction route produces boron with lower purity (90–97%). From the commercial point of view, the gaseous reaction is more expensive, whereas the magnesiothermic reaction produces amorphous B with a purity value of 95–97% at a much lower cost [25].

In this study, the B powder was purchased from a local company. The purity on the tag is "99.9% metals basis" [26]. However, the B powder comes from the magnesiothermic

route, which reduces $B_2O_3$ using Mg. The primary impurities in this route could be $MgB_2$, MgO, and $B_2O_3$. MgO, which has high stability and high melting point, can remain inert in the heat treatment method. Considering that MgO was produced together with B, the MgO could be embedded into the B particles instead of only on the surface. Therefore, we surmised that it might be challenging to eliminate the oxygen in the B powder coming from the magnesiothermic reaction route.

*What is the effect of the purification of B powder on the superconducting properties of $MgB_2$?*

After systematically investigating the effect of the purification of B powder on the superconducting connectivity, it can be found that even only a partial purification of the B powder can significantly affect the superconducting properties of bulk $MgB_2$. The $J_c$ at the low field of the sample with purified B power is improved; however, pinning is reduced simultaneously. Further, notably, the pinning center became smaller according to our fitting. The superconducting connectivity was doubled. Even the B powder was only partially purified.

From the results, an initial conclusion is that the MgO could be the pinning center to $MgB_2$ and a supercurrent barrier. When the B powder is purified, the MgO amount decreases. Consequently, the pinning force at the high field is decreased, and the superconducting connectivity is enhanced. There is another speculation that the purified B powder may also cause a reduction in the size of MgO, which expresses a change in the pinning mechanism.

Moreover, the purification of the B powder can help us further understand the superconducting properties of pure polycrystalline $MgB_2$ without oxygen, thereby allowing us to reveal the effect of MgO finally. As a future study, the superconducting connectivity that significantly increased due to eliminating MgO sized B powder from the gaseous reaction route, such as that from Specialty Materials Inc., USA [27], or Pavezyum Company, Turkey [28], could be purified and investigated.

## 5. Conclusions

In this study, we investigated the effect of particle size on the purification results for B powder. The particle size reduction can aid the chemical method toward purifying the B powder further but does not affect the heat treatment method. The effect of the initial purifying effect on the superconducting properties was also investigated. The purification of the B powder can eliminate the MgO impurities in the bulk $MgB_2$. $J_c$ of the purified sample was much higher at the low field but dropped faster when the magnetic field increased. The superconducting connectivity significantly increased due to the elimination of oxygen, probably MgO. It can be concluded that the purification of B powder is the key to understanding the superconducting properties of pure $MgB_2$ and the effect of MgO. We also discussed the reasons why oxygen could not be eliminated.

**Author Contributions:** L.Y. contributes to the sample preparation and most of the experiment. The L.M., M.L., W.Z. contribute to the measurement of PPMS and TEM. The Z.Z. contributes to the supervision of the whole experiment and original writing. The L.W., J.L. and Q.W. contribute to the supervision of the investigation and writing—review and editing. H.S. project administration. All authors have read and agreed to the published version of the manuscript.

**Funding:** Z.Z. was supported by the Magnetic Resonance Union of Chinese Academy of Sciences (2020GZL001). H.S. was supported by the Beijing Municipal Natural Science Foundation (2212025), L.W. was supported by the National Natural Science Foundation of China (51807191). J.L. was supported by the National Natural Science Foundation of China (51777205). Q.W. was supported by the National Natural Science Foundation (12042506).

**Institutional Review Board Statement:** Not applicable.

**Informed Consent Statement:** Not applicable.

**Data Availability Statement:** Data sharing not applicable.

**Conflicts of Interest:** The authors declare no conflict of interest.

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
