# Peer review of "Exploration of the Effect of Oxygen on Superconductivity in MgB2 Bulk by Using Boron Powder with Different Particle and Purification"

_crystals, doi:10.3390/cryst11030278_

Round 1

Reviewer 1 Report

In " Exploration of the effect of oxygen on superconductivity in 2
MgB2 bulk by using boron powder with different particle and 3
purification" the author used two methods- chemical and heat for purification of boron powder. The aim is to remove the oxygen content, however, based on the experimental results this aim is fulfilled only partially. Although, the oxygen content is reduced only partially, the effect on superconducting properties is noticeable and worth reporting in crystals. I recommend the publication in crystals after the author address following questions.

  1. Based on pinning force phenomenon, author itself explains that with B2O3 reduction, the possible MgO content is also reduced after purification. If so then why author relate the enhanced superconducting properties only to reduced content of B2O3? enhanced superconducting properties can also be related to reduced content of MgO.
  2. For only partial removal of oxygen content author doubt on the purchased quality of Boron powder, however, it is less convincing because the claimed quality is 99.9%. did author try some other boron powder and compare the results?
  3. Author should further explains about which method-heat or chemical is better in removing overall oxygen or enhancing the superconducting properties? may be author can elaborate the advantages and disadvantages of each method.

Author Response

Reviewer 1

In " Exploration of the effect of oxygen on superconductivity in MgB2 bulk by using boron powder with different particle and purification" the author used two methods- chemical and heat for purification of boron powder. The aim is to remove the oxygen content, however, based on the experimental results this aim is fulfilled only partially. Although, the oxygen content is reduced only partially, the effect on superconducting properties is noticeable and worth reporting in crystals. I recommend the publication in crystals after the author address following questions.

We appreciate the excellent suggestion from reviewer. Based on the three questions, we made substaintial change on the discussion section of this paper.

Q1. Based on pinning force phenomenon, author itself explains that with B2O3 reduction, the possible MgO content is also reduced after purification. If so then why author relate the enhanced superconducting properties only to reduced content of B2O3? enhanced superconducting properties can also be related to reduced content of MgO.

A1: Yes, we agree that both the improvement of the superconducting connectivity and the decreasing of the pinning are coming from reducing MgO content. We thought that the main reason for the reduction of MgO is coming from the reduction of B2O3. However, this conculsion is not hundreds percent sure. So we think the reviewer’s suggestion is correct, the change of superconducting properties is coming from the reduction of MgO. We already changed the relevant description.

Q2. For only partial removal of oxygen content author doubt on the purchased quality of Boron powder, however, it is less convincing because the claimed quality is 99.9%. did author try some other boron powder and compare the results?

A2: I agree with the reviewer’s comment. It is only our speculation why we couldn’t further purify the oxygen in boron powder. It is not a hundred perscent sure, so we already changed our description.

In this paper, the boron powder is coming from Aladdin. Recently, we also tried the micro size boron powder Macklin and Alfa Aesar. Further, nanosize boron powder from Pavezyum Company is also under testing. The purification results is very different.

The microsize boron powder from Alfa Aesar can get much better purification results than Aladdin. The boron powder's oxygen content can decrease from around 30000 to less than 5000 after the boron powder was ground and purified by heat treatment. However, the boron powder from Macklin shows completely different results. The initial oxygen content is more than 40000, and we couldn’t decrease it to less than 15000. Even more, the Jc value of MgB2 bulk using the powder from Macklin is very low, and we don’t know the reason yet.

For the nanosize one, the initial oxygen content is only around 4000, and for purification, the oxygen content is around 1700.

From the results above, the particle size affects the oxygen content and purification effect of boron powder. But we also thought there are some other factors to affect the purification of boron powder.

Q3. Author should further explains about which method-heat or chemical is better in removing overall oxygen or enhancing the superconducting properties? may be author can elaborate the advantages and disadvantages of each method.

A3: We already added a paragraph to discuss the advantage and disadvantage of each method in the discussion part.

Reviewer 2 Report

Yang et al. have investigated the effect of oxygen on MgB2 bulk. Authors have performed experiments and analyzed data well. The role of MgO is still not understandable so far. I would recommend its publication after minor revision.

1) How did authors measured R-T of particles, It is not mentioned in methods.

2) Are M-T measured in zero field ?

Author Response

Reviewer 2

Yang et al. have investigated the effect of oxygen on MgB2 bulk. Authors have performed experiments and analyzed data well. The role of MgO is still not understandable so far. I would recommend its publication after minor revision.

Q1) How did authors measured R-T of particles, It is not mentioned in methods.

A1: I am sorry for the misleading. The R-T curve was measured on the MgB2 bulk after synthesized instead of the powder. We already changed the relevant description.

Q2) Are M-T measured in zero field ?

A2: The M-T curve was measured at 50 Oe (5mT) applied field. We already changed the relevant description.

Round 2

Reviewer 1 Report

recommended for publication